# Treatment of Metastatic Melanoma with a Combination of Immunotherapies and Molecularly Targeted Therapies

**DOI:** 10.3390/cancers14153779

**Published:** 2022-08-03

**Authors:** Taylor Rager, Adam Eckburg, Meet Patel, Rong Qiu, Shahina Gantiwala, Katrina Dovalovsky, Kelly Fan, Katie Lam, Claire Roesler, Aayush Rastogi, Shruti Gautam, Namrata Dube, Bridget Morgan, S M Nasifuzzaman, Dhruv Ramaswami, Varun Gnanasekar, Jeffrey Smith, Aftab Merchant, Neelu Puri

**Affiliations:** 1College of Medicine, University of Illinois, Rockford, IL 61107, USA; trager2@uic.edu (T.R.); mpate307@uic.edu (M.P.); rqiu5@uic.edu (R.Q.); sganti3@uic.edu (S.G.); dovalov2@uic.edu (K.D.); kfan20@uic.edu (K.F.); klam23@uic.edu (K.L.); croesl2@uic.edu (C.R.); arasto5@uic.edu (A.R.); sgauta3@uic.edu (S.G.); ndube3@uic.edu (N.D.); bmorga7@uic.edu (B.M.); snasif3@uic.edu (S.M.N.); sanram75@gmail.com (D.R.); theepiclegogamer30@gmail.com (V.G.); jsmith59@uic.edu (J.S.); amerc@uic.edu (A.M.); 2Feinberg School of Medicine, Northwestern University, Chicago, IL 60611, USA; adam-eckburg@northwestern.edu

**Keywords:** molecularly targeted therapies, immunotherapies, melanoma, clinical trials, drug resistance, mutations

## Abstract

**Simple Summary:**

Immunotherapies and molecularly targeted therapies have drastically changed the therapeutic approach for unresectable advanced or metastatic melanoma. The majority of melanoma patients have benefitted from these therapies; however, some patients acquire resistance to them. Novel combinations of immunotherapies and molecularly targeted therapies may be more efficient in treating these patients. In this review, we discuss various combination therapies under pre-clinical and clinical development which can reduce toxicity, enhance efficacy, and prevent recurrences in patients with metastatic melanoma.

**Abstract:**

Melanoma possesses invasive metastatic growth patterns and is one of the most aggressive types of skin cancer. In 2021, it is estimated that 7180 deaths were attributed to melanoma in the United States alone. Once melanoma metastasizes, traditional therapies are no longer effective. Instead, immunotherapies, such as ipilimumab, pembrolizumab, and nivolumab, are the treatment options for malignant melanoma. Several biomarkers involved in tumorigenesis have been identified as potential targets for molecularly targeted melanoma therapy, such as tyrosine kinase inhibitors (TKIs). Unfortunately, melanoma quickly acquires resistance to these molecularly targeted therapies. To bypass resistance, combination treatment with immunotherapies and single or multiple TKIs have been employed and have been shown to improve the prognosis of melanoma patients compared to monotherapy. This review discusses several combination therapies that target melanoma biomarkers, such as BRAF, MEK, RAS, c-KIT, VEGFR, c-MET and PI3K. Several of these regimens are already FDA-approved for treating metastatic melanoma, while others are still in clinical trials. Continued research into the causes of resistance and factors influencing the efficacy of these combination treatments, such as specific mutations in oncogenic proteins, may further improve the effectiveness of combination therapies, providing a better prognosis for melanoma patients.

## 1. Introduction

In 2021, an estimated 106,110 people in the United States were diagnosed with cutaneous melanoma, of which 7180 died [1,2]. Melanoma accounts for only 1% of skin cancers, but more people die from this disease than any other type of skin cancer [3]. Melanoma develops due to exposure to UV or radiation-induced DNA damage, which triggers the malignant transformation of melanocytes [4]. Most melanomas have mutations in the RAS/RAF/MAPK and PI3K/AKT cellular regulation pathways [5,6]. In addition, mutations in upstream proteins that are involved in the pathways and factors that cause vascularization, have been implicated in melanoma pathogenesis [7]. While dacarbazine and surgical resection can treat early manifestations of melanoma, these therapeutic options are ineffective in metastatic melanoma, resulting in low survival rates in patients [8]. Immunotherapies such as ipilimumab, nivolumab, and pembrolizumab currently play a vital role in treating malignant melanoma [9]. Although these agents have shown improved efficacy and lower toxicity than previously used cytokine-based treatments, more research is needed to develop new immunotherapies for patients with late-stage melanoma [10].

Molecularly targeted tyrosine kinase inhibitors (TKIs) that act on specific components of the RAS/RAF/MAPK and PI3K/AKT pathways, have efficacy in treatment of metastatic melanoma. However, resistance to these agents develops when they are used as monotherapy for extended periods of time [5,11]. Furthermore, the phenomenon of epithelial–mesenchymal transition and the upregulation of numerous oncogenic proteins have been implicated in resistance development [12]. Nevertheless, combining TKIs and immunotherapies targeting key proteins in the oncogenic RAS/RAF/MAPK or PI3K/AKT pathways has been shown to delay the onset of resistance, resulting in improved progression-free survival in patients with metastatic melanoma (Table 1) [13,14].

TKIs have been further combined with targeted immunotherapeutic agents or agents targeting the VEGFR-mediated vascularization pathway in a number of trials with noteworthy efficacy [47]. In this review, we will comprehensively elucidate the current state of targeted combination therapies for advanced melanoma. The therapeutic effectiveness and combinatory potential demonstrated in recent pre-clinical and clinical trials of agents targeting BRAF, MEK, NRAS, KRAS, HRAS, c-Kit, c-MET, VEGFR, and PI3K/AKT will be discussed in detail (Figure 1). The summarization of the efficacy of current combination therapies makes this review an important addition to the field of melanoma therapy (Table 2).

## 2. BRAF

BRAF serine/threonine kinase (BRAF) is a cellular regulator of proliferation and survival that is mutated in more than 60% of cutaneous melanomas [48]. The BRAF V600E missense mutation is a valine-to-glutamic acid substitution that accounts for more than 80% of all BRAF mutations, while V600K and V600D accounts for the majority of the remaining 20% [48]. Molecularly targeted therapies to target BRAF mutant melanomas are thus gaining clinical interest and significance [49]. Several melanoma therapies targeting constitutively activated BRAF have been investigated in recent years, with several approaching therapeutic approval [49]. Both single agent and combination therapies that combine BRAF inhibitors with complementary agents such as MEK inhibitors or immunotherapy have proven effective and will be discussed in this review. It is common for resistance to emerge soon after the initiation of these therapies. Since BRAF is an important protein in the RAS/RAF/MAPK signaling pathway, resistance develops after prolonged treatment with BRAF inhibitor by reactivating or bypassing the MAPK pathway [49]. Therefore, treatment regimens that inhibit BRAF signaling while preventing acquired resistance are also gaining importance in the field of molecularly targeted melanoma therapy. 

Vemurafenib, dabrafenib, and encorafenib are three drugs used to treat BRAF V600E-mutated melanoma. These drugs have demonstrated clinical promise when used alone or in combination with multiple agents. In 2011, vemurafenib became the first FDA-approved drug to target melanoma with V600E mutations in BRAF [50]. In phase III clinical trials, comparison of vemurafenib with dacarbazine for the treatment of BRAF V600E-mutated metastatic melanoma showed that vemurafenib was associated with a 74% reduction in the risk of death or disease progression [51]. Since vemurafenib inhibits the MAPK pathway, combining it with a MEK inhibitor such as cobimetinib, may enhance the effects and improve overall and progression-free survival in patients. In a randomized, double-blinded phase III clinical trial involving 495 individuals with BRAF V600-mutated, unresectable late-stage melanoma, the combination of vemurafenib and cobimetinib was associated with a median progression-free survival of 12.3 months, compared to 7.2 months in those who received vemurafenib and a placebo [52]. Although the combination regimen had a higher toxicity profile than vemurafenib and a placebo, the safety profile of the combination was tolerable and manageable, indicating its status as a superior first-line therapeutic [52,53]. In 2015, the FDA approved using the combination of vemurafenib and cobimetinib in the treatment of unresectable melanoma [54]. Finally, the addition of immunotherapeutic options such as atezolizumab, a PD-L1 monoclonal antibody, to established therapeutic cocktails has gained the interest of researchers in recent years [55] (Figure 2). A recent phase I/II open-label clinical trial by Louveau et al. investigated the use of fixed-dose Vemurafenib with Palbociclib, a CDK4/6 inhibitor. No significant drug-drug interactions were highlighted, and a clinical response was measured in 5 of 18 enrolled patients. Furthermore, the median progression-free survival was 2.8 months. These results are a promising foundation on which future studies assessing BRAFi and CDK4/6i combination therapy could be built [56].

Although immunotherapy is associated with lower overall response rates (ORR) in patients with BRAF-mutated melanoma, using checkpoint inhibitors in combination with BRAF inhibitors and MEK inhibitors may provide a more durable response that is less susceptible to spontaneous resistance [57]. The recently published IMspire150 phase III trial by Genentech, Inc., Exelixis, and Roche investigated the effect of a three-part combination of atezolizumab, vemurafenib, and cobimetinib in advanced BRAF-mutated melanoma [18]. The group that received all three agents had 15.1 months of progression-free survival as compared to 10.6 months in the control group that was treated with vemurafenib, cobimetinib, and a placebo [18]. Blood creatinine phosphokinase, alanine aminotransferase, and lipase levels increased in the treatment group but not significantly compared to the control group [18]. Consequently, the triple combination of an immune checkpoint inhibitor (ICIs), BRAF inhibitor, and MEK inhibitor proved to be a more effective, safe, and tolerable first-line therapy than earlier combination regimens [18]. On 30 July 2020, the FDA approved this triple combination for therapeutic use in BRAF V600-mutated melanoma.

Dabrafenib is another BRAF inhibitor approved by the FDA in 2013 with a long history of use as both a solo first-line therapy in BRAF V600E-mutated late-stage melanoma as well as in combination regimens. Similar to Vemurafenib, Dabrafenib has been shown to have an improved response rate, overall survival (OS), and progression-free survival (PFS) when compared to dacarbazine [50,58]. Dabrafenib, combined with trametinib, a MEK inhibitor, is one of the main therapeutic options. The efficacy of this combination as a first-line therapy was compared to vemurafenib monotherapy in an open-label, phase III clinical trial of patients with either BRAF V600E-mutated or BRAF V600K-mutated advanced melanoma [59]. Dabrafenib plus trametinib resulted in 11.4 months of progression-free survival with no additional toxicity versus 7.3 months in the monotherapy group. [59]. In a separate study, 84 patients with BRAF V600-mutated unresectable melanoma were started on dabrafenib alone and then switched to a dabrefenib and trametinib combination before disease progression. Of those 84 individuals, 22 benefited from the combination therapy without sacrificing any safety [60]. In 2014, this combination was approved by the FDA for use in metastatic and unresectable BRAF V600E/K-mutated melanoma, and in 2018, this approval was extended for use in stage III melanoma [54]. This combination has recently been used in conjunction with another therapeutic PD-1 blockade agent, pembrolizumab, in phase II clinical trials [15,16,61]. Although promising results were seen, including increased progression-free survival and improved responsiveness to the regimen, toxicity and adverse events also increased significantly [15,16,61]. This study paved the way for future research on triple therapies with dabrefenib, a MEK inhibitor, and a checkpoint inhibitor.

Encorafenib, the most recent BRAF inhibitor for advanced V600-mutated melanoma, was approved by the FDA in June 2018 [50]. Recent phase III clinical trial findings showed that encorafenib monotherapy was more effective than vemurafenib in the treatment of BRAF-mutated metastatic melanoma [62,63]. Subsequent trial data analyzed encorafenib in combination with binimetinib, a MEK inhibitor, and found that this combination improved overall survival rates in melanoma patients compared to both encorafenib and vemurafenib monotherapies [62,63]. Specifically, the addition of binimetinib to the regimen allows encorafenib to be delivered in higher doses, increasing response rates to a greater extent. However, this has also been associated with an increase in adverse events [62,63,64]. Unlike vemurafenib and dabrafenib and their combination counterparts, encorafenib was immediately approved by the FDA for use with binimetinib [50,62]. This combination of encorafenib and binimetinib has recently been tested in conjunction with Palbociclib in the CELEBRATE trial which is in the early clinical stage [65]. 

The long-term clinical efficacy of the therapies mentioned above is highly dependent on the time from treatment initiation until the development of resistance. Several therapies that specifically target BRAF-inhibitor-resistant phenotypes have been studied. Recently, two unique paradox inhibitors, PLX8394 and PLX7904, were described in colon adenocarcinoma and resistant melanoma xenografts due to BRAF amplification [49]. These therapies work without stimulating MAPK and have been shown to induce apoptosis ex vivo with more efficacy than vemurafenib alone. However, these results did not correlate with in vivo studies due to acquired resistance by activation of ERK1/2 through heterogeneous mechanisms [66,67]. Future experiments should focus on paradox inhibitors, specifically in unresectable BRAF V600E-mutant melanoma, to better understand their therapeutic potential. Microphthalmia-associated transcription factor (MITF) amplification has been associated with resistance to BRAF inhibitors by altering the MAPK pathway. The combination of paradox inhibitors with suppressors of MITF amplification have shown promise for treating resistant late-stage melanoma [49]. CH5552074 and CH6869398 are two MITF protein suppressors that have shown in vivo efficacy in melanoma xenograft models [68].

Several controversies remain in the field of BRAF-mutated melanoma treatment. Studies have found that 10% of melanoma patients harboring a BRAF mutation have the V600K driver mutation, the second most common type behind V600E. Therapeutic strategies in the context of V600K BRAF-mutated melanoma are not as well elucidated as those with V600E and prognosis is poor. Melanoma driven by V600K mutations is thought to have unique features compared to V600E, including decreased reliance on MAPK/ERK pathway over activation, increased expression of c-KIT, and upregulation of the PIK3CB-AKT anti-apoptotic pathway. Although treatment with BRAF/MEK combinatory regimens such as dabrafenib + trametinib, vemurafenib + cobimetinib, and encorafenib + binimetinib, as well as with immunomodulators such as pembrolizumab and iplimumab, have shown some efficacy in V600K-mutant melanoma, evidence is not strong enough to make any formal first-line treatment recommendation for melanoma patients with the V600K mutation. Moving forward, the BRAF V600K driver mutation should be independently investigated in greater detail to formulate specified and evidence-backed recommendations of treatments for patients with this mutation, as their prognosis is currently still poor [69].

## 3. MEK

Most patients diagnosed with melanoma have mutations in their oncogenes which regulate signaling through the mitogen activated protein kinase (MAPK) pathway [70]. Such patients can be screened and categorized for the type of mutations found in oncogenes and tumor suppressor genes involved in the MAPK pathway [70]. Selective inhibitors of MAPK pathway mediators, such as RAF, ERK, and MEK, can effectively diminish and treat melanoma when used in combination therapy [70]. Patients with metastatic BRAF V600 mutant melanoma, which accounts for 40–50% of melanoma patients, are treated with MEK inhibitors such as trametinib, cobimetinib, selumetinib, and binimetinib [70]. Using MEK inhibitors alone, or after treatment with BRAF inhibitors, was found to be ineffective [71]. Preclinical studies also demonstrated that acquired resistance to BRAFi can be due to MAPK pathway and blocking this pathway may induce apoptosis in BRAF V600 mutant melanoma [72]. The downstream inhibition of MEK1/2 in combination with BRAFi, is used to inhibit MAPK pathway and prevent emergence of resistance [54]. As previously discussed, numerous combinations of BRAF/MEK inhibitors such as dabrafenib/trametinib and vemurafenib/cobimetinib, have proven to be more effective than BRAF inhibitors alone in treating BRAF V600 mutant melanoma [70]. Clinical studies suggest that combining BRAF and MEK inhibitors impede drug resistance more than BRAF inhibitors alone and also minimize the risk of recurrence of skin diseases such as cutaneous squamous cell cancer (cSCC) [14,73]. 

Several clinical trials are currently investigating combinations of immune checkpoint blockades, specifically anti-PD-1/L1 therapies, with BRAF and/or MEK inhibitors [74]. Recent studies suggest that inhibition of BRAF and MEK, combined with anti-PD-1/L1 antibodies, improves prognosis of melanoma patients in a CD8 T cell dependent mechanism [74]. Long-term studies are needed to determine the risks and toxicities associated with each agent when used in combination [74].

Other combination therapy clinical trials, such as COMBI-i, have investigated MEK inhibitors plus cyclin-dependent-kinase (CDK) 4/6 inhibitors, or MEK inhibitor plus an anti-PD-1/PD-L1 antibody. Specifically, these researchers looked at spartalizumab (an anti-PD-1 antibody), in combination with dabrafenib and trametinib, versus placebo plus dabrafenib and trametinib in patients with BRAF V600-mutant unresectable or metastatic melanoma. The pre-clinical trial showed promise for improving treatment outcomes and progression-free survival (PFS), however, this study is currently inconclusive. Furthermore, researchers speculate that a subpopulation with specific biomarkers may benefit from this combination therapy in the future [17]. In the COMBI-AD trial, stage III patients had a longer disease-free survival (DFS) when treated with a combination of dabrafenib and trametinib [75,76]. In another study, patients with surgically removed mucosal melanoma were treated with, high-dose interferon (IFN)-alfa, or chemotherapy with a combination of temozolomide and cisplatin. After receiving chemotherapy, patients showed a significantly longer DFS and OS than those treated with high-dose IFN-alfa or the ones in the observation group [14]. Furthermore, recent preclinical work by Nassar et al. found that the combination of Trametinib and Palbociclib significantly decreased tumor growth in vivo in xenograft models that were resistant to both BRAF and MEK inhibitors. However, in vitro, the same combination downregulated cellular growth, upregulated cell-cycle arrest, and inhibited downstream MAPK signaling in cell lines resistant to BRAF inhibitors. This new study is significant because it demonstrated that combined CDK4/6 knockout with MEK inhibition has the potential to overcome the acquisition of BRAF/MEK inhibitor resistance, which is thought to be primarily mediated through MAPK reactivation [77]. 

IFNs were often used as an adjuvant therapy before the advent of novel systemic therapies for treatment of melanoma. High-dose IFN-alpha improved DFS, however, changes in OS were marginal, and several adverse effects were observed [14]. Therefore, the data presented thus far suggest that more research is needed on combination and adjuvant therapies for the effective treatment of melanoma. 

## 4. NRAS

NRAS is commonly mutated in several cancers, and it serves as an important oncogenic driver in melanoma [78]. Mutations at codon 61, and less frequently at codons 12 and 13, account for 15–20% of all melanoma cases. Ras substitution mutations often result in impaired GTPase activity and inactivation of the Ras GTPase-activating protein, both of which stabilize the active, GTP-associated conformation of Ras. Melanoma with a high NRAS burden is associated with thicker tumors, higher mitotic activity, and a worse prognosis [79,80,81]. Patients with advanced, surgically incurable NRAS-mutant melanoma are treated with immunotherapy (including IL1, ipilimumab, pembrolizumab, and nivolumab) as first-line therapy and chemotherapy as second-line therapy [82,83].

To date, no NRAS-specific targeted therapy has been approved, though various inhibitors targeting the MAPK pathway and several combination therapies have been evaluated for NRAS-mutant melanoma. For example, MEK inhibitors in combination with other inhibitors are examined frequently. A preclinical study showed that simultaneous inhibition of both MEK, which is downstream of Ras, and CDK4/6, can induce tumor regression in NRAS-mutant melanoma. In a phase Ib/II, open-label clinical trial, 102 patients with locally advanced or metastatic NRAS-mutant melanoma were treated with a MEK inhibitor (binimetinib), and a CDK4/6 inhibitor (ribociclib). All patients reported adverse events, the most common being elevated creatinine phosphokinase level, diarrhea, nausea, and fatigue. The pharmacokinetic profile suggested the absence of drug–drug interaction [84,85]. Among the 41 patients enrolled in the phase II dose expansion trial, the median progression-free survival was 3.7 months, and the overall survival was 11.3 months. Among trial participants, 20% achieved a partial response, 51% had stable disease, and 15% had progressive disease [22]. To our knowledge, this is the first and the largest phase II trial completed on combination therapy in patients with NRAS-mutant melanoma.

Melanoma cell lines with NRAS mutations at codon 61 were treated with Amgen Compd A (a pan-RAF inhibitor) and trametinib (a MEK inhibitor) to synergistically suppress cell growth. Cell lines sensitive to this combination expressed more phosphorylated MEK, indicating that cell proliferation was highly dependent on the MAPK pathway. Cyclin D1, which is regulated by the MAPK pathway and interacts with CDK4/6 to induce G1-to-S phase transition, might be an important intermediate for cell growth in these sensitive cell lines. This study found that inhibiting RAF and MEK could effectively control cell growth in NRAS-mutant melanoma cell lines that are dependent on the MAPK pathway. It also demonstrates a need to target alternative pathways because resistant cell lines may use a pro-survival pathway independent of MAPK [19]. 

To address drug resistance to MEK inhibitors, recent preclinical studies evaluated combination therapies that may help overcome resistance to MEKi. A study showed that co-inhibition of MEK and ERK5, a compensatory pathway activated by MEKi possibly via receptor tyrosine kinase PDGFRβ, suppressed growth and progression of NRAS-mutant melanoma cells in vitro and in xenografts [23]. Additionally, inhibition of PDPK1 synergistically enhanced the efficacy of MEKi by stimulating antitumor immunity [24]. Another study suggested that suppression of the Rho/myocardin-related transcription factor (MRTF) pathway, which is activated in MEKi resistance, with CCG-222740 enhanced the cytotoxicity of trametinib in NRAS-mutant cells [25]. An alternative strategy for overcoming MEKi resistance is by inhibiting PHGDH, an enzyme involved in the serine synthesis pathway, that is upregulated in MEKi-resistant cells [26]. Furthermore, CD147/VEGFR-2 peptide inhibitor reduced malignant potential through the STAT3 pathway and resensitized MEKi-resistant NRAS-mutant xenografts to MEKi. The MEKi-CD147i combination induced tumor regression in PDX [27]. These strategies, though in their early phase of investigation, may help overcome resistance and enhance the efficacy of inhibitors targeting the MAPK pathway.

Since immunotherapy is the first-line treatment for NRAS-mutant melanoma, a team of researchers conducted a case-control study to retrospectively analyze 236 patients with NRAS-mutant melanoma and 128 patients with wild-type NRAS melanoma who had received checkpoint blockade therapies, such as ipilimumab (an anti-CTLA-4 antibody) and anti-PD-1 therapy. Patients with NRAS mutations had a significantly shorter median overall survival (21 months) than those with wild-type NRAS (33 months). Among patients with NRAS mutations, those who had received MEK inhibitors, such as binimetinib, pimasertib, or trametinib, in addition to immunotherapy, had a five-month longer median overall survival (25 months) compared to those who did not receive a MEK inhibitor (20 months) [20]. This study suggests that MEK inhibitors may enhance the efficacy of immunotherapy; however, a randomized clinical trial is still warranted.

In addition to inhibiting the MAPK pathway with MEK inhibitors, suppression of targets upstream of Ras has also been studied in patients with NRAS-mutant melanoma. In animal models, combining tivantinib (an inhibitor of c-MET receptor tyrosine kinase) with sorafenib (an inhibitor of VEGFR, PDGFR, and RAF kinases) led to a synergistic antiproliferative effect. In a subsequent phase I trial, ten patients with NRAS-mutant melanoma, one patient with wild type NRAS melanoma, and eight melanoma patients of unknown NRAS status received both tivantinib and sorafenib. Patients with NRAS mutations had a lower overall response rate compared to patients with wild type or unknown NRAS status. However, patients with mutated NRAS treated with combination therapy had a longer median progression-free survival. Although these results were not statistically significant, the well-tolerated toxicity profile of this combination therapy suggested its potential application for slowing down resistance to either inhibitor [21,86].

Several ongoing clinical trials are investigating the use of combination therapies for metastatic melanoma. A phase Ib, multicenter, open-label trial is currently evaluating the safety profile of belvarafenib (a pan-RAF inhibitor) as a single agent or in combination with either cobimetinib or cobimetinib plus atezolizumab (PD-L1 inhibitor) in patients with NRAS-mutant melanoma previously treated with anti-PD-1/PD-L1. An estimate of 98 patients will be assigned into these three treatment regimens. A dosing phase followed by an expansion phase will be conducted in the two-agent combination arm, while a run-in phase followed by an expansion phase will be completed in the three-agent arm. This trial is expected to be completed in November 2024. Similarly, another open label, phase Ib trial focuses on the combination effect of cobimetinib and IN10018 (an adenosine triphosphate–competitive focal adhesion kinase inhibitor) in NRAS-mutant melanoma and metastatic uveal melanoma. A total of 52 participants will be assigned to either the IN10018 monotherapy arm or the combination treatment arm. This clinical trial will be completed in June 2023. In addition, an open label, phase Ib clinical trial is examining naporafenib (LXH254, a RAF inhibitor) in combination with either LTT462 (an ERK1/2 inhibitor), trametinib, or ribociclib (a CDK4/6 inhibitor) in NRAS or BRAFV600 mutant melanoma and non-small cell lung cancer. In total, 241 patients have enrolled in this trial and have been non-randomly assigned to either of these three arms. Concurrently, a phase II trial with about 320 participants randomly assigned to these arms is underway and is expected to be completed by September 2023. Lastly, in another phase I trial with 29 patients, trametinib and hydroxychloroquine (an autophagy inhibitor) in NRAS-mutant melanoma and neuroblastoma are being investigated. Preclinical studies show that suppression of Ras downstream signaling induces autophagy that may result in resistance to BRAF and MEK inhibitors. Inhibition of both MEK and autophagy resulted in tumor regression in patient-derived xenografts of NRAS-mutated melanoma [87]. The current clinical trial is investigating the dosing and tolerability profile of co-inhibiting MEK and autophagy.

Thus far, no combination therapy for NRAS-mutant melanoma has progressed to phase III clinical trials. Most of the completed and ongoing clinical trials involving combination therapy for advanced NRAS-mutant melanoma target signaling molecules upstream or downstream of Ras, such as MEK and RAF. Other studies are investigating the combination of MAPK inhibition with immunotherapy or inhibitors that target alternative pathways involved in cell cycle progression and autophagy.

## 5. HRAS and KRAS

KRAS is among the most commonly mutated oncogenes in human cancer. It encodes a GTPase in the RAS/MAPK pathway, promoting cell growth and proliferation. KRAS mutations occur almost exclusively at the G12 position and are most often glycine-to-cysteine. These point mutations favor the active conformation of the KRAS protein (GTP-bound). Increased KRAS activity leads to uninhibited cell growth and proliferation in cancer. KRAS inhibitors are primarily being studied in the context of non-small cell lung cancer and colorectal cancer; however, some clinical trials are exploring the use of these therapies in other advanced or metastatic solid tumors. KRAS mutations are found in about 1% of melanomas [88].

HRAS is the least frequently mutated GTPase among the three Ras isoforms, and its transcriptional regulation differs from that of NRAS and KRAS. HRAS mutations are present in only 1.5% of all melanoma cases, and elevated HRAS expression is correlated with shorter survival in patients with cutaneous melanomas [89,90]. HRAS is not a typical mutation marker used in a clinical setting for melanoma patients, and its role in solid tumors has received less attention. One preclinical study evaluated the efficacy of ASN007, an ERK1/2 inhibitor, in cell lines with mutated RAS, including HRAS. ASN007 has potent antitumor activity, and this effect was enhanced by the PI3K inhibitor copanlisib [28].

## 6. c-KIT

c-KIT is a proto-oncogene that encodes the type III transmembrane receptor tyrosine kinase. c-KIT is responsible for the activation of various downstream signaling cascades, including the MAPK/ERK, PI3K/AKT, and JAK/STAT pathways, resulting in alterations of gene expression for cell survival, proliferation, differentiation, and apoptosis [91,92]. Changes in c-KIT can thus play a role in various types of cancer, influencing metastasis, tumor growth, and cell proliferation. c-KIT mutations are associated with gastrointestinal stromal tumors (GIST), leukemia, lung cancer, and melanoma [93,94,95,96]. In melanoma, the most common c-KIT mutation is in exon 11, while other mutations are found in exons 9, 13, and 17 [97]. Furthermore, KIT mutations are positively associated with older age, chronic sun damage, as well as mucosal and acral melanomas [98]. KIT mutations are present in 3% to 9.5% of melanoma patients; mutations are present in approximately 23% of acral melanomas and 15.6% of mucosal melanomas. With an increase in the copy number of KIT, these numbers increase to about 36% and 39%, respectively [98,99,100,101,102].

Imatinib, a c-abl, bcr-abl, and platelet-derived growth-factor receptor (PDGFR) tyrosine kinase inhibitor, has been studied for its effectiveness in inhibiting c-KIT in a variety of malignancies, including human mast cell leukemia and GIST. It has been shown to have promising therapeutic effects in metastatic melanoma [103,104,105]. While imatinib improves melanoma patients with KIT alterations, subsequent KIT mutations result in imatinib resistance [106]. Nilotinib, is a c-KIT inhibitor that is used to treat patients with chronic myeloid leukemia, and is beneficial for overcoming resistance development to imatinib [92,107]. Several other c-KIT inhibitors such as dasatinib, sunitinib, sorafenib, and masitinib have been studied as future therapeutic agents for melanoma; however, further studies are needed to evaluate their clinical efficacy. 

Several clinical trials are studying the efficacy of c-KIT inhibitors together with other TKIs or immunotherapies as c-KIT mutant melanoma patients often develop resistance to TKIs [108]. Previous clinical trials have shown the effectiveness of antibody therapy targeting PD-1 or its ligand in melanoma, providing an opportunity for further research into combination therapy [109]. In a case report published in 2019, imatinib, a KIT inhibitor, was combined with pembrolizumab, a PD-1 inhibitor, in a patient with metastatic melanoma with double-mutant KIT receptor [29]. The patient was given imatinib 400 mg daily and pembrolizumab 200 mg every three weeks. After 3 months of treatment, there was a significant decrease in the size of the metastatic melanoma, and the patient reached full remission with no evidence of metastasis after 6 months of therapy. After about 12 months of continued imatinib and pembrolizumab therapy, the melanoma recurred, with eventual metastasis to the brain. This case report illustrates the transitory effectiveness of combination therapy with the KIT inhibitor imatinib and immunotherapy. Another study assessed therapy with imatinib and NN2101-DM1, a conjugated IgG1 and microtubule inhibitor, both together and alone in c-KIT mutant cancers in mouse models. GIST treated with this combination therapy achieved remission compared to either therapy alone which highlights a potential avenue of research for combination therapy in melanoma management [110]. A phase I clinical trial explored the efficacy of the KIT inhibitor dasatinib with the alkylating agent dacarbazine in the treatment of metastatic melanoma [30]. Each of the five cohorts received a different combination of dasatinib and dacarbazine dosages. Four of the 29 patients in the cohort exhibited a partial response to 70 mg of dasatinib received twice daily, and 17 had stable disease with treatment. In total, 26 out of 46 patients across all cohorts had stable disease, with a median progression-free survival of 13.4 weeks and a median overall survival of 40.6 weeks. This response was better than that of monotherapy with dasatinib, which previously resulted in a progression-free survival of 8 weeks, and monotherapy with dacarbazine, which did not prevent tumor growth. Overall, the combination therapy of a KIT inhibitor and immunotherapy offers promise in the treatment of melanoma with KIT alterations. A case report describes a patient with stage IV M1b metastatic and mucosal melanoma with a KIT mutation who attained a complete response with combination therapy consisting of sorafenib, temozolomide, wide local excision, and radiation [31]. Sorafenib has demonstrated improved efficacy in melanoma treatment when combined with paclitaxel and carboplatin [32]. However, clinical trials that demonstrated efficacy of sorafenib in combination therapy were not specific to melanoma with KIT alterations. In another study, patients are being recruited to determine the efficacy of combinations of nilotinib with dabrafenib/trametinib in metastatic melanoma patients with BRAF mutations [NCT04903119]. While clinical trials support the potential benefit of combination therapy in the treatment of melanoma, more clinical trials are necessary to fully understand the role of KIT inhibitors in combination with immunotherapy. Furthermore, a phase II clinical trial of nilotinib in 25 patients with unresectable melanoma with KIT mutations demonstrated a response rate of 20% and a disease control rate of 56% in patients with exon 11 or 13 mutations after receiving 400 mg oral nilotinib twice daily for 6 months. Of note, the data from this study also showed that STAT3 expression in good responders was significantly decreased, while levels in poor responders did not show significant change. While this clinical trial did not assess outcomes of nilotinib in combination with other therapies, the study showed a significant association between reduced STAT3 signaling and a clinical response [92]. This study suggests that the JAK/STAT pathway may mediate response to nilotinib and indicates a need to further evaluate the potential for use of JAK/STAT inhibitors in combination with KIT inhibitors in the treatment of KIT-mutated melanoma. Evaluation of combination therapies using KIT inhibitors and STAT inhibitors for treatment of melanoma is thus far minimally studied and therefore demands further consideration.

## 7. VEGFR

Vascular endothelial growth factor (VEGF) activates vascular endothelial growth factor receptors (VEGFRs) and is involved in both normal and pathological angiogenesis [111]. VEGFRs exhibit pro-angiogenic activity through the RAS/RAF/MAPK signaling pathway and are also responsible for increasing vascular permeability and promoting cell migration [111,112]. Recent research has shown that higher VEGF-A expression correlates with later pathological stages of melanoma [113]. Anti-VEGF drugs such as axitinib and bevacizumab inhibit tumor angiogenesis and are used to treat certain advanced cancers. Axitinib is used to treat renal cell carcinoma, and bevacizumab is used to treat cervical cancer [114,115]. Interestingly, a study released in 2019 showed that bevacizumab may be contraindicated for uveal melanoma because of its effect in reducing oxidative stress-induced cell death [116]. No VEGFR inhibitors are currently FDA-approved for treating melanoma. VEGF inhibitors have a limited effect on the overall survival of cancer patients; thus, efforts have shifted toward employing these drugs in combination therapy [117,118].

The primary area of interest surrounding VEGFR inhibitors is their effectiveness when administered in combination with other drugs, such as ICIs. Immune checkpoint inhibitors, such as PD-1/PD-L1 and CTLA-4 inhibitors, increase immune-mediated destruction of tumor cells by interrupting tumor inhibition of T-cell-mediated apoptosis [119]. The effectiveness of ICIs, however, is dependent on the tumor microenvironment, of which tumor-associated macrophages are a major component [120]. According to a study on hepatocellular carcinoma, VEGF inhibitors may shift the tumor microenvironment from immunosuppressive to immunostimulatory, which responds to immune checkpoint blockade [121,122,123]. In a preclinical study on cutaneous melanoma in murine models, the anti-VEGFR-1 mAb D16F7 increased the ratio of M1 to M2 macrophages and CD8+ to FOXP3+ cells, favoring an antitumor environment. When applied in combination with the CTLA-4 inhibitor Ipilimumab and anti-PD-1 mAbs, it produced more effective inhibition of melanoma growth compared to the ICI alone [124]. The combination of bevacizumab and ipilimumab is under phase II investigation in patients with metastatic melanoma after phase I results showed manageable toxicities and an improved median overall survival of 25.1 months compared to 10.1 months with ipilimumab alone [36,125].

Several phase II and phase III clinical trials are currently investigating the combination of pembrolizumab, a PD-1 inhibitor, and lenvatinib, a VEGFR inhibitor, in advanced melanoma. These endeavors were initiated by favorable results in a phase Ib/II study revealing an acceptable safety profile and 48% ORR [38,39]. Another phase Ib trial showed clinical activity of PD-1 inhibitor toripalimab in combination with axitinib in patients with mucosal melanoma [35]. Another phase II clinical trial is recruiting advanced mucosal melanoma patients after a phase Ib trial showed acceptable outcomes and possible anti-tumor activity after treatment with toripalimab plus axitinib [35]. A third, newer anti-PD-1 antibody, which is named SHR-1210 or camrelizumab, is being studied in combination with apatinib in patients with various cancers, including metastatic acral melanoma [33].

An alternative approach to combine antineoplastics in conjunction with VEGFR therapies is being investigated. A combination of bevacizumab with carboplatin and paclitaxel was studied in a randomized phase II trial in patients with advanced mucosal melanoma. Results showed the median PFS of 4.8 months and the median OS of 13.6 months, which is significantly longer than the carboplatin/paclitaxel arm that showed a median PFS of 3.0 months and a median OS of 9.0 months [37]. A phase II trial combining axitinib with paclitaxel and carboplatin found a median PFS of 8.7 months, which was longer than the median PFS of approved targeted therapies such as vemurafenib with a median PFS of 5.3 months, dabrafenib with a median PFS of 5.1 months, and ipilimumab with a median PFS of less than 3.0 months [126]. In addition, pazopanib, a multi-kinase inhibitor of VEGFR-1, 2, and 3, PDGFR, and c-KIT, is being combined with paclitaxel to treat patients with metastatic melanoma [40]. This combination was well-tolerated and had anti-tumor activity equivalent to current first-line treatments such as ICIs or a BRAF/MEK TKIs [40]. A dose-escalation study on advanced melanoma refractory to targeted therapy investigated the combination of apatinib and temozolomide, another cytotoxic drug. This combination regimen yielded 8.3% objective response rate and 83% disease control rate [34].

Unlike the other VEGFR therapies mentioned thus far, aflibercept, a chimeric recombinant protein that serves as a novel decoy VEGF receptor, is being investigated as an independent treatment, as well as in combination. A phase II study of ziv-aflibercept combined with interleukin-2 (IL-2) in patients with unresectable stage III/IV melanoma found an increase in PFS from 2.3 to 6.9 months [127]. A phase I trial is currently recruiting patients with advanced solid tumors, including melanoma, to study the combination of pembrolizumab and ziv-aflibercept (Table 2). The trials examining VEGFR therapies primarily assessed their ability to enhance the efficacy of existing melanoma targeted, immunotherapies, and chemotherapies rather than as a solo treatment. Acquired resistance to common melanoma treatments, such as BRAF inhibitors, is inevitable [128]. Preclinical data suggest that VEGFR-1 upregulation may contribute to melanoma progression once cells acquire resistance to vemurafenib [129]. 

Future studies should investigate the therapeutic potential of a VEGFR inhibitor to treat BRAF-resistant melanoma; however, antiangiogenic administration should be used with caution. Small studies on intravitreal injections of bevacizumab in patients with uveal melanoma and patients with choroidal melanoma resulted in increased tumor growth [130,131]. This phenomenon was also observed after cessation of bevacizumab in colorectal cancer patients, resulting in increased proangiogenic factors and cancer stem cells [132]. Studies also suggest that there is a correlation between ICI therapy, particularly anti-PD-1/PD-L1 antibodies, and hyper-progressive disease in a subset of patients with solid tumors [133]. Another study on melanoma patients treated with ICIs, however, considered hyper-progression as a relatively rare phenomenon [134]. These conflicting findings should be considered as a potential consequence when combining these therapies. The evaluation of VEGFR combination therapies is still in its early stages, with no completed phase III or IV trials. Initial results are promising and indicate that VEGFR inhibitors might be useful as adjunctive therapies.

## 8. c-MET

c-MET is a tyrosine kinase receptor (RTK) encoded by the MET proto-oncogene. It binds to the ligand hepatocyte growth factor (HGF), activating the PI3K-AKT, STAT, SRC, and MAPK signaling pathways, resulting in tumor cell proliferation, invasion, and metastasis [135,136,137,138]. Hence, c-MET is a crucial target as its downregulation silences many connected pathways involved in tumorigenicity [139]. As HGF is produced only by the mesenchymal cells when required, it typically interacts with its receptor, c-MET, in a paracrine fashion in skin cells. In melanoma cells, there is sustained autocrine production of HGF resulting in activation of the c-MET receptor that is no longer confined to mesenchymal cells [140]. Our earlier paper described how c-MET could serve as a potential target against metastatic melanoma. With several FDA-approved inhibitors against c-MET, the future possibilities of using it in combination therapies appears to be vast [49].

Currently, several tyrosine kinase inhibitors (TKI) targeting c-MET are being studied. However, the resistance to these inhibitors has reduced their effectiveness against melanoma and the mechanism of resistance needs to be further elucidated [49]. SU11274, a c-MET TKI, was linked to an increase in phosphorylation of c-MET, mTOR, p7056Kinase, and active β-catenin and GATA-6 upregulation. Upon acquiring resistance, the m-TOR pathway and Wnt signaling were found to be upregulated, which may cause resistance to c-MET inhibitors. Combining mTOR inhibitor, everolimus, and Wnt inhibitor, XAV939, with SU11274 resulted in a 95% reduction in tumors in melanoma cell lines [41].

Tivantinib is a c-MET TKI inhibitor, and sorafenib is a BRAF, c-KIT, and FLT3 multi-kinase inhibitor. These two drugs have been successfully studied individually in phase III clinical trials. However, they are now being studied in combination in a phase I clinical trial. This study includes 19 patients, and is focused on the safety, tolerability, and effective dosage of the combination therapy. In addition, 10 out of the 19 patients have NRAS-mutated melanoma, a mutation that makes the tumor more pharmacologically susceptible to c-MET inhibition. The response rate in these cases was found to be 26%, whereas the response rate in the disease control group was 63% [21].

Recent studies have shown that it is possible to produce the same efficacy of monoclonal antibodies in a bispecific manner. A study used a bi-specific antibody (BsAb) that targets two crucial oncogenic factors in cancer: c-MET and PD-1. This antibody binds and blocks c-MET, disrupting HGF/c-MET-mediated signaling and causes apoptosis. Moreover, it has a high affinity for blocking PD-1, a T cell receptor which, when coupled with its ligand PD-L1, suppresses immunity against cancer. Blocking PD-1 on T cells boosts the body’s immunity against cancer cells that often overexpress PD-L1 [44]. Cabozantinib, a MET inhibitor that targets several tyrosine kinase receptors, such as VEGFR, c-MET, and AXL is being studied in conjunction with pembrolizumab, a humanized immunoglobulin that acts against PD-1, this combination is also being studied in a phase I/II clinical trial in patients with advanced metastatic melanoma (NCT03957551). 

Uveal melanoma (UM) is the most common eye cancer, accounting for 5% of all melanomas [141]. In UM cell cultures, HGF was found to be one of the drivers of this resistance [142], and treating UM with a combination of MEK and c-Met inhibitors could be an effective combination therapy. Recent in vitro studies have shown that LY2801653, a type II kinase inhibitor targeting c-Met, contributes to overcoming HGF-mediated resistance in UM cell lines treated with trametinib, a MEK1/2 inhibitor. A significant decrease (64–81%) in cancer cell survival rate was reported [42].

Furthermore, MEK inhibitors in combination with AKT inhibitors, PI3K inhibitors, or c-MET inhibitors can potentially improve the clinical efficacy of these drugs in metastatic uveal melanoma. However, MEK-based therapies are generally toxic and cannot be used for long-term treatment. To circumvent this barrier, cyclin-dependent kinase (CDK) 4/6 inhibition was investigated. A recent study found that combining abemaciclib, a CDK4/6 inhibitor, with merestinib, a c-MET inhibitor, reduced FOXM1 expression, and significantly inhibited tumor growth in metastatic UM tumors grown in human HGF-knock-in immunocompromised mouse models [43].

## 9. PI3K/AKT

Although the phosphatidylinositol 3-kinase (PI3K)/AKT pathway is frequently involved in melanomas, mutations of PI3K itself are rare. Rather, the pathway is activated in other ways. One cause is via a loss-of-functional mutation in the NF1 tumor suppressor gene, which occurs in 10–15% of melanoma cases [143]. As a result, the NRAS protein becomes hyperactive, causing the MAPK and PI3K pathways to be activated [143]. Another way the PI3K/AKT pathway becomes activated is due to a BRAF mutation coupled with a loss-of-function mutation in the PTEN gene [143], activating not only the PI3K pathway, but also the MAPK pathway [143]. This occurs in about 10–30% of melanoma cases [144]. Given that the PI3K-AKT pathway is a regulator of many cellular processes and also a pathway frequently activated in cancer [143], inhibition of this pathway is a promising target for pharmacological therapy.

Numerous PI3K inhibitors are currently under investigation in clinical trials. One research study aimed to determine the maximum tolerated dose (MTD) of two experimental drugs, pimasertib (MSC1936369B, mitogen-activated protein extracellular signal-regulated kinase (MEK) inhibitor) and voxtalisib (SAR245409, a PI3K inhibitor), administered in combination. Both drugs were administered orally with increasing dosages in 21-day cycles until MTD was attained. The MTD in this study was defined as intolerable toxicity, the investigator’s decision to discontinue treatment, or withdrawal of consent by the subject. After reaching MTD, participants were separated into tumor-specific expansion cohorts: breast cancer, non-small cell lung cancer (NSCLC), melanoma, and colorectal cancer. One of the fifteen melanoma patients in the study had a complete tumor response to the drugs; one had a partial response; four had progressive disease; seven had stable disease; and one was not evaluated [45] [NCT01390818].

Another ongoing study is exploring the efficacy of BKM120 (buparlisib—PI3K inhibitor) in combination with LGX818/MEK162 (BRAF inhibitor/MEK inhibitor) in treating BRAF V600 melanoma. This study includes 160 melanoma patients and is likely to be completed by January 2023. In the first phase of the study, participants were given LGX818/MEK162 until disease progression. LGX818/MEK162 will then be combined with one of the following drugs in phase II of the study: LEE011 (ribociclib—selective CDK4/6 inhibitor), BGJ398 (infigratinib—FGFR kinase inhibitor), BKM120, or INC280 (capmatinib—inhibitor). Genetic analysis will determine which drug each participant will receive. After combination therapy is initiated, study participants will be followed over the course of two years to determine the overall response rate (ORR) for each drug combination.

Another PI3K inhibitor under investigation is the selective PI3K-beta inhibitor GSK2636771. Combination therapy of GSK2636771 with pembrolizumab is being studied as a treatment for patients with metastatic melanoma and PTEN loss who failed to respond to other forms of treatment. There are currently 36 patients enrolled in this study. In phase 1 of this study, the MTD of GSK2636771 will first be determined. The first group of patients will be given 300 mg of GSK2636771 for 21 days, with the dose increasing with each successive group of new patients. The MTD from phase I will be utilized to evaluate the overall response rate of the combination therapy in phase II of the study. In both phases, patients will receive 200 mg of IV Pembrolizumab every three weeks. Patients will be routinely monitored for any adverse side effects or drug toxicities. The estimated completion date for this trial is December 2022. 

Lastly, an ongoing trial is evaluating the safety of the PI3K-gamma inhibitor IPI-549 as monotherapy as well as IPI-549 in combination with nivolumab. Parts A–C of the study are concerned with determining dose-limiting toxicities (DLT). Participants in Part A of the study are given IPI-549 orally once a day until disease progression. Then, in Part B, the dose is increased to twice a day until disease progression. Part C involves participants receiving IPI-549 orally at doses determined in Parts A and B, along with a nivolumab IV infusion every two weeks. Part D focuses on IPI-549 monotherapy, whose dose is determined again by Parts A and B. Participants with melanoma are enrolled in Part E of the study, in which they receive IPI-549 orally at a dose determined from Parts A/B/C, in combination with nivolumab IV infusion every two weeks. This study was supposed to be completed in June 2021; however, no findings have been released yet. 

Thus far, we have solely discussed PI3K inhibitors undergoing clinical trials; however, therapies targeting AKT, which is closely related to PI3K, are also being studied. AKT inhibitors prevent the phosphorylation of mTOR, thereby exhibiting profound effects on cell survival, proliferation, and angiogenesis. As such, they are a useful target for therapy, and multiple AKT inhibitors are currently undergoing clinical trials. Ipatasertib (GDC-0068), an AKT inhibitor, is being studied in combination with atezolizumab. Participants are given increasing doses of ipatasertib in combination with a fixed dose of atezolizumab. Phase I of this study aims to determine the maximum tolerated dose of the combined drugs. Phase II aims to determine the number and type of treatment-related adverse events associated with the two-drug combination. Participants are separated into six cohorts according to tumor types. A total of 12 melanoma patients have agreed to participate in this study [NCT03673787].

Another AKT inhibitor in clinical trials is MK-2206. It is being investigated in combination with other targeted therapies to treat solid tumors, including melanoma. In a phase I clinical trial, MK-2206 was studied in combination with hydroxychloroquine to treat advanced solid tumors. All 62 patients enrolled in this study had melanoma, renal cell cancer, or prostate cancer. Phase I of the study, which assessed the maximum tolerated dose (MTD), was completed on 14 February 2020, but no findings have been released to date. In addition, MK-2206 combined with carboplatin/paclitaxel, docetaxel, or erlotinib was also explored in a phase I clinical trial to treat solid tumors. Of the 72 patients enrolled in this study six had melanoma, and of those six patients, only one showed a partial response to the combination therapy. The combination therapy was generally well-tolerated, with the most common adverse effects being a maculopapular rash and febrile neutropenia [46].

## 10. Combination Therapies with Immune Checkpoint Inhibitors

Immunotherapy, which blocks immune checkpoints, has been shown to induce longer-lasting responses in about one-third of cancer patients [145]. The five-year survival rate of patients with advanced melanoma treated with a combination of immunotherapies is now approaching 50% [146]. Identifying new checkpoint targets can further improve the outcomes of these therapies [146]. Programmed cell death protein 1 (PD1) is a T cell receptor that binds to its ligand (PD-L1) found on antigen-presenting cells (APCs) and tumor cells to inhibit T cell activation [147]. Recent studies focus on combining PD1/PD-L1 antibodies with anti-CTLA-4 inhibitors to target various cancers, including melanoma. The principle for using this combination is that inhibition of the PD-1/PD-L1 pathway alone does not induce antitumor immunity because cancer cells lack antigen-specific CD8-positive T cells. Additional inhibition of the CTLA-4 pathway, on the other hand, causes increased activation of CD8-positive cells in lymph nodes and tumor cells [148] (Figure 3).

The efficacy of immune-checkpoint inhibitors, such as ipilimumab (anti-CTLA-4 antibody), nivolumab, and pembrolizumab (anti-PD-1 antibodies), outperforms conventional therapies in clinical trials [14]. Several phase II and III trials showed that combining nivolumab with ipilimumab increased response rates up to 50–60% [149]. A randomized phase II trial of nivolumab plus ipilimumab compared to ipilimumab (2:1) showed an OS of 63.8% for the patients treated with the combination compared to 53.6% for patients treated with ipilimumab alone [149]. KEYNOTE-006 randomized phase III trial showed that pembrolizumab has a significantly longer PFS and OS than ipilimumab [150,151]. Additionally, administration of ipilimumab after pembrolizumab and nivolumab therapy demonstrated improved outcomes compared to treatment in the opposite sequence in the CheckMate 064 randomized phase II trial [152]. In the CheckMate 067 randomized trial, nivolumab alone or in combination with ipilimumab was found to be better than ipilimumab monotherapy [153,154]. In this trial, 945 patients with unresectable stage III or IV melanoma received ipilimumab combined with nivolumab or monotherapy with nivolumab or ipilimumab alone. The combination demonstrated three- and four-year OS rates of 58% and 53%, compared to 52% and 46% for nivolumab and 34% and 30% for ipilimumab alone. In patients with PD-L1-positive tumors, the median PFS was 14.0 months in both nivolumab-plus-ipilimumab and the nivolumab groups. However, in PD-L1-negative patients, PFS was 11.2 months with the combination therapy compared to 5.3 months with nivolumab alone [155]. Based on these favorable outcomes, nivolumab plus ipilimumab was the first immunotherapy combination approved by the FDA in 2015 [10].

Lymphocyte-activation gene 3 (LAG-3) is a co-receptor found on activated T cells such as CD4+ and CD8+ T cells, and Tregs [156]. LAG-3 is upregulated in many cancers, including melanoma, and is also responsible for regulating the immune checkpoint pathway, which inhibits T-cell activity. Relatlimab, is an IgG4 monoclonal LAG-3 inhibitor that restores the functions of T cells [157]. Relatimab combined with nivolumab has been shown in clinical trials to enhance antitumor immune responses in melanoma patients [158]. RELATIVITY-047, a phase II/III, global, double-blind, randomized trial, compared the combination treatment of relatimab plus nivolumab to nivolumab alone in patients with advanced melanoma [158]. The median PFS for patients administered with combination treatment was 10.1 months, compared to 4.6 months in patients treated with nivolumab alone. Furthermore, after 12 months of initiation, the PFS of patients treated with the combination was 47.7%, compared to 36% in patients treated with nivolumab alone [87]. Based on these encouraging findings, in March 2022, the FDA approved the combination of nivolumab and rilatimab for use in patients 12 years of age or older with unresectable or metastatic melanoma.

Several clinical trials are investigating the combination of immunotherapies with radiation therapies [159] to increase the efficacy of immune checkpoint inhibitors. Other immunotherapies involve using viruses, such as oncolytic viruses, which are injected directly into the tumor to induce the release of tumor-associated antigens and mediate the host’s antitumor response [14]. For example, in the OPTiM phase III trial, talimogene laherparepvec (T-VEC), a genetically modified herpes simplex virus-1, had a 16% durable response rate [160]. Furthermore, studies are also focusing on targeting other immune checkpoints such as TIM3 (T cell immunoglobulin and mucin domain-containing protein 3) and TIGIT (T cell immune receptor with Ig and ITIM domains) in combination with already approved ICIs to improve treatment efficacy in melanoma patients [159].

## 11. Conclusions

In this review, we discussed several promising combinations of targeted TKIs and immunotherapies for treating malignant melanoma. We also extensively discussed therapies targeting mutations in the oncogenic RAS/RAF/MAPK and PI3K/AKT pathways and the VEGFR, c-Kit, c-Met-mediated vascularization, and downstream dysregulation pathways. Many of these therapies have shown to be clinically effective in combination with one another or with immunotherapeutic agents. In addition, the newly designed combination therapies described can potentially combat TKI resistance, a critical therapeutic issue in utilizing targeted genetic therapies [5,11]. However, continued research into factors such as exon mutations that affect the efficacy and toxicity profiles of these treatments is essential. Furthermore, future research in combination-targeted therapy is necessary to improve the survival of patients with late-stage melanoma.

## Figures and Tables

**Figure 1 cancers-14-03779-f001:**
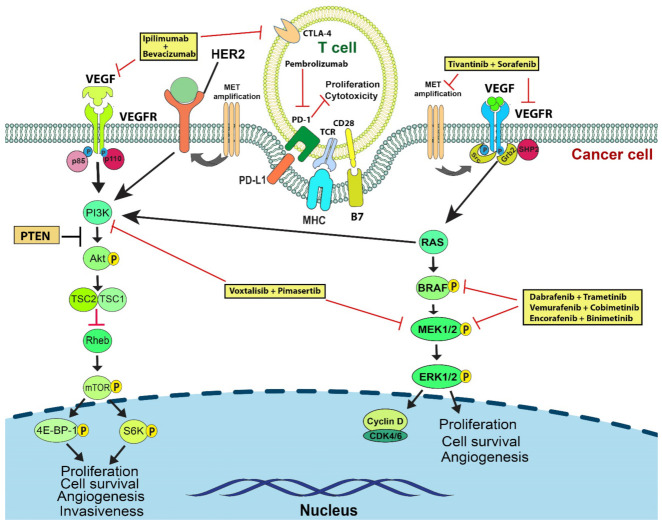
Combination therapies targeting T cell and the RAS/RAF/MAPK and PI3K/AKT pathways which regulate cell proliferation in melanoma. Growth factors, such as VEGF, bind to their receptors and activate receptor tyrosine kinase (RTK), which phosphorylates Grb2, which then forms a complex with SOS. SOS induces a conformational change in Ras to favor Ras binding to GTP. GTP-bound Ras activates the Ras/MEK/ERK and PI3K/AKT/mTOR pathways, promoting tumorigenesis, angiogenesis, and cell motility. One of the downstream effects of Ras activation is the expression of cyclin D, which activates CDK4/6 and allows for progression of the cell cycle. Furthermore, melanoma cells employ multiple ways to evade immunosurveillance, for example, through the PD-L1expression. The induction of PD-L1 with the PD-1 receptor on T cells suppresses T cell proliferation and effector functions. Combination therapies (yellow boxes) targeting different parts of these pathways may exert synergistic anti-tumor effects or prevent the development of resistance to monotherapy.

**Figure 2 cancers-14-03779-f002:**
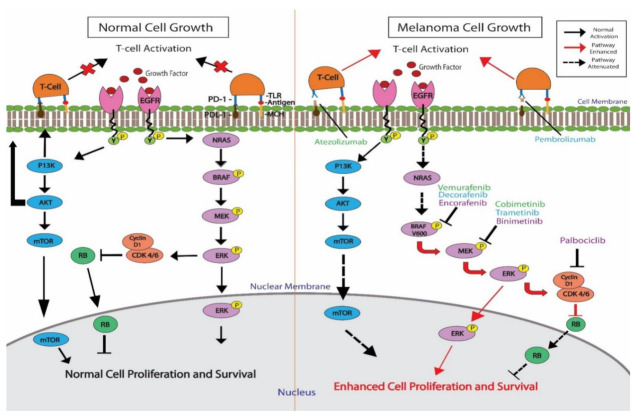
Combination therapies and their respective targets in BRAF-mutated melanoma. In melanoma, BRAF upregulation leads to persistent upregulation of downstream targets MEK and CDK 4/6. T-cell activation is also enhanced in these cancer states. In this figure, the different mechanistic targets of three triple therapies are depicted, with each triple therapy subunit written in the same color, either green, blue, or purple. The three regimens shown are atezolizumab/vemurafenib/cobimetinib, decorafenib/trametinib/pembrolizumab, and encorafenib/binimetinib/palbociclib. Each combination consists of a BRAF inhibitor, MEK inhibitor, and either a checkpoint inhibitor or a CDK 4/6 blockade.

**Figure 3 cancers-14-03779-f003:**
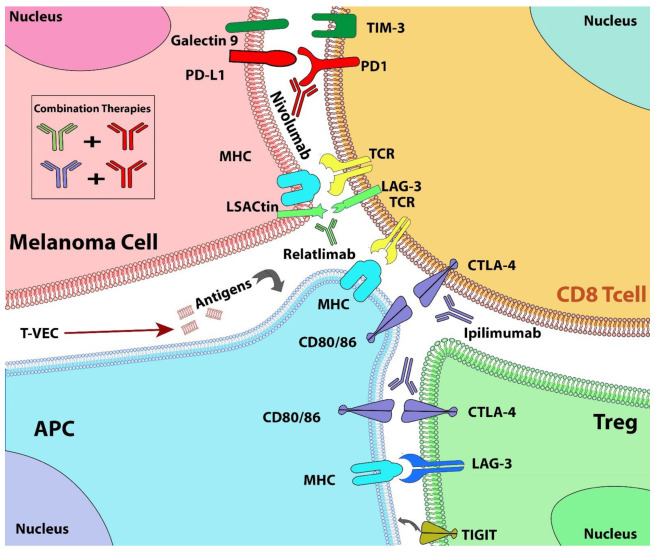
Combination of various immune checkpoint inhibitors. Combinations of immunotherapies, such as PD-1/PD-L1 antibodies, e.g., nivolumab and anti-CTLA-4 inhibitors, e.g., ipilimumab, have been shown to improve survival rates in melanoma patients. The addition of ipilimumab to the nivolumab regimen increases the infiltration of CD8-positive cells in the tumor. Another recently approved combination of LAG3 inhibitor relatlimab with nivolumab increases activation of CD4+ and CD8+ T cells, and Tregs, resulting in antitumor responses.

**Table 1 cancers-14-03779-t001:** Combination therapies for melanoma.

Drug 1 (Target)	Drug 2 (Target)	Drug 3 (Target)	Type of Study	Reference
**BRAF/MEK**				
Dabrafenib (BRAF)	Trametinib (MEK)	Pembrolizumab (PD-1)	Phase IIPhase I/II	[15,16]
Dabrafenib (BRAF)	Trametinib (MEK)	Spartalizumab (PD-1)	Phase III	[17]
Vemurafenib (BRAF)	Cobimetinib (MEK)	Atezolizumab (PD-L1)	Phase III	[18]
**NRAS**				
Compd A (RAF)	Trametinib (MEK)		Pre-clinical	[19]
Ipilimumab (CTLA-4) or anti-PD-1 (PD-1)	Binimetinib, Pimasertinib, or Trametinib (MEK)		Case control studies	[20]
Tivantinib (cMET)	Sorafenib (VEGFR/PDGFR/RAF/other)		Phase I	[21]
Binimetinib (MEK)	Ribociclib (CDK4/6)		Phase Ib/II	[22]
Trametinib (MEK)	XMD9-92 (ERK5)		Pre-clinical	[23]
Trametinib (MEK)	GSK2334470 (PDPK1)		Pre-clinical	[24]
Trametinib (MEK)	CCG-222740 (MRTF)		Pre-clinical	[25]
Trametinib or PD901 (MEK)	PHGDH siRNA		Pre-clinical	[26]
Cobimetinib (MEK)	CD147 inhibitor		Pre-clinical	[27]
**HRAS**				
ASN007 (ERK1/2)	Copanlisib (PI3K)		Pre-clinical	[28]
**KIT**				
Imatinib (KIT/other)	Pembrolizumab (PD-1)		Case Report	[29]
Dasatinib (KIT)	Dacarbazine		Phase 1	[30]
Sorafenib (KIT/other)	Temozolomide		Case Report	[31]
Sorafenib (KIT)	Carboplatin	Paclitaxel	Phase I/II	[32]
**VEGFR**				
Apatinib (VEGFR)	Camrelizumab/SHR-1210 (PD-1)		Phase II/III	[33]
Apatinib (VEGFR)	Temozolomide (Antineoplastic)		Clinical trial (escalation study)	[34]
Axitinib (VEGFR)	Toripalimab (PD-1)		Phase Ib	[35]
Bevacizumab (VEGFR)	Ipilimumab (CTLA-4)		Phase I	[36]
Bevacizumab (VEGFR)	Paclitaxel (Antineoplastic)	Carboplatin (Antineoplastic)	Phase II	[37]
Lenvatinib (VEGFR)	Pembrolizumab (PD-1)		Phase Ib/II	[38,39]
Pazopanib (VEGFR/PDGFR/c-KIT)	Paclitaxel (Antineoplastic)		Phase II	[40]
**C-MET**				
Everolimus (mTOR)	XAV939 (Wnt)	SU11274 (cMET)	Pre-clinical	[41]
LY2801653 (cMET)	Trametinib (MEK1/2)		Pre-clinical	[42]
Abemaciclib (CDK4/6)	Merestinib (cMET)		Pre-clinical	[43]
Bi-Specific antibody (cMET & PD1)			Pre-clinical	[44]
**PI3K/AKT**				
Pimasertib (MEK1/2)	Voxtalisib (pan-PI3K)		Phase Ib	[45]
MK-2206 (AKT)	Carboplatin/Paclitaxel, Docetaxel, or Erlotinib		Phase I	[46]

**Table 2 cancers-14-03779-t002:** Current combination therapy clinical trials for advanced/metastatic melanoma.

Clinical Trials	Phase/Status	Participants	Conditions	Drug Intervention (Drug Target)	Primary Outcome Measures	Estimated Completion Date
NCT04720768 (CELEBRATE)	Ib, Recruiting	78	Metastatic BRAF Mutant Melanoma	Encorafenib (BRAF) + Binimetinib (MEK) + Palbociclib (CDK4/6)	Dose-Limiting Toxicity	12/04/2023
NCT04835805	Ib, Recruiting	98	Advanced NRAS Mutant Melanoma, Had received anti-PD-1/PD-L1 therapy	Belvarafenib (RAF) + Cobimetinib (MEK) with/without Atezolizumab (PD-L1)	Dose-Limiting Toxicity, Adverse Events	11/11/2024
NCT04109456	Ib, Recruiting	52	Metastatic Uveal Melanoma, Metastatic NRAS Mutant Melanoma	IN10018 (FAK) + Cobimetinib (MEK)	Safety, Tolerability	06/30/2023
NCT02974725	Ib, Active, not recruiting	241	Metastatic/Advanced KRAS or BRAF Mutant Non-Small Cell Lung Cancer or NRAS Mutant Melanoma	Naporafenib (RAF) + LTT462 (ERK1/2)/Trametinib (MEK)/Ribociclib (CDK4/6)	Adverse Events, Dose-Limiting Toxicities, Tolerability	11/25/2022
NCT04417621	II, Recruiting	320	Previously Treated Unresectable or Metastatic BRAFV600 or NRAS Mutant Melanoma	Naporafenib (RAF) + LTT462 (ERK1/2)/Trametinib (MEK)/Ribociclib (CDK4/6)	Overall Response Rate	09/08/2023
NCT03979651 (CHLOROTRAMMEL)	I, Recruiting	29	Metastatic/Advanced NRAS Melanoma	Trametinib (MEK) + Hydroxychloroquine (autophagy)	Dose-Limiting Toxicities, Partial or Complete response	03/31/2022
NCT04903119	I, Recruiting	15	Metastatic or Unresectable melanoma with BRAF V600	Nilotinib (cKIT) + Dabrafenib (BRAF) + Trametinib (MEK)	Dose-Limiting Toxicities	03/31/2027
NCT02298959	I, Recruiting	78	Advanced Solid Tumors	Aflibercept (VEGFR) + Pembrolizumab (PD-1)	Safety, Recommended Phase II Dosing	11/31/2022
NCT02159066 (LOGIC-2)	II, Active, not recruiting	160	Locally Advanced or Metastatic BRAF V600 Melanoma	Encorafenib (BRAF) + Binimetinib (MEK) + Ribociclib (CDK4/6)/Infigratinib (FGFR kinase)/Buparlisib (PI3K)/ Capmatinib (MET)	Overall Response Rate	01/17/2023
NCT03957551	Ib/II, Recruiting	39	Advanced Melanoma	Cabozantinib (VEGFR/cMET/AXL) + Pembrolizumab (PD-1)	Dose-Limiting Toxicities, Overall Response Rate	07/01/2024
NCT03131908	I/II, Active, not recruiting	36	Refractory Metastatic Melanoma with loss of PTEN	GSK2636771 (PI3Kβ) + Pembrolizumab (PD-1)	Maximum Tolerated Dose, Objective Response Rate	12/31/2022
NCT02637531	I, Active, not recruiting	219	Advanced Melanoma	IPI-549 (PI3K-gamma) + Nivolumab (PD-1)	Dose-limiting Toxicity, Adverse Events	12/2022
NCT03673787 (IceCAP)	I/II, Recruiting	87	Solid Tumors with Hyperactive PI3K	Ipatasertib (AKT) + Atezolizumab (PD-L1)	Maximum Tolerated Dose, Adverse Events	11/2023
NCT01480154	I, Active, not recruiting	62	Advanced Solid Tumors	MK-2206 (AKT) + Hydroxychloroquine	Maximum Tolerated Dose, Dose-Limiting Rate	02/14/2020, not published
NCT03470922 (RELATIVITY-047)	II/III, Active, not recruiting	714	Previously Untreated Metastatic or Unresectable Melanoma	Relatlimab (LAG-3) + Nivolumab (PD-1)	Progression Free Survival	11/30/2023

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
