# Peer review of "Treatment of Metastatic Melanoma with a Combination of Immunotherapies and Molecularly Targeted Therapies"

_cancers, 2022, doi:10.3390/cancers14153779_

Round 1

Reviewer 1 Report

The review article entitled “Treatment of metastatic melanoma with a combination of immunotherapies and molecularly targeted therapies” provides a comprehensive overview on targets for molecularly targeted and immunotherapies for melanoma patients and highlights avenues for combination therapies. The review is well structured and well written.  The table and figures summarize the discussed topics well. However, a few details in the table and the figures should be checked and corrected.

Figure 1: The depiction of VEGF inhibition with Bevacizumab + Ipilimumab is misleading and looks like Ipilimumab would inhibit VEGF.

Figure 3: the depiction of antibodies to indicate immune checkpoint blockade inside the cell might also be misleading and looks like the neutralizing antibodies bind at intracellular domains. 

Table 1: Cobimetinib is labeled as BRAF inhibitor, though it is a MEK inhibitor. Please check. Imatinib is listed as a KIT inhibitor. This might also be misleading and it should at least be labeled as KIT and others. 

Abstract: The numbers provided for melanoma patients and melanoma-related deaths only account for the US. This is not clear in the abstract.

Author Response

To the Editor of Cancers,                                                                                                

Subject: Submission of revised manuscript No. cancers-1748189: “Treatment of metastatic melanoma with a combination of immunotherapies and molecularly targeted therapies.”

Dear Editor,

We would like to thank the editors and reviewers for their valuable comments and suggestions that have helped in improving the quality of this manuscript. Responses to each comment are provided below. We are confident that this revised manuscript will now be acceptable for publication in Cancers.

Answers to reviewer comments:

The review article entitled “Treatment of metastatic melanoma with a combination of immunotherapies and molecularly targeted therapies” provides a comprehensive overview on targets for molecularly targeted and immunotherapies for melanoma patients and highlights avenues for combination therapies. The review is well structured and well written.  The table and figures summarize the discussed topics well. However, a few details in the table and the figures should be checked and corrected.

Comment; Figure 1: The depiction of VEGF inhibition with Bevacizumab + Ipilimumab is misleading and looks like Ipilimumab would inhibit VEGF.

Response – We appreciate valuable feedback from the reviewer, we updated the figure 1 to depict that Ipilimumab inhibits CTLA-4.

Comment; Figure 3: the depiction of antibodies to indicate immune checkpoint blockade inside the cell might also be misleading and looks like the neutralizing antibodies bind at intracellular domains.

Response – We moved the immune checkpoint inhibitors outside the cell so that it doesn’t seem like neutralizing antibodies bind at intracellular domains.

Comment; Table 1: Cobimetinib is labeled as BRAF inhibitor, though it is a MEK inhibitor. Please check. Imatinib is listed as a KIT inhibitor. This might also be misleading and it should at least be labeled as KIT and others.

Response - We updated the information in the table regarding Cobimetinib and Imatinib and have highlighted it in yellow.

Comment; Abstract: The numbers provided for melanoma patients and melanoma-related deaths only account for the US. This is not clear in the abstract.

Response – We have added additional clarification in the abstract that these statics apply to the United States.

We hope that these answers suffice the reviewer’s concerns and that the revised manuscript may now be accepted for publication in Cancers.

This manuscript is original and is not under consideration elsewhere. None of the manuscript contents have been previously published. All authors have read and approved all versions of the manuscript and its content for the submission to Cancers.

Sincerely,

Neelu Puri Ph.D. (Corresponding author)Associate ProfessorUniversity of Illinois College of MedicineDept. of Biomedical Sciences1601 Parkview AveRockford, IL 61107Telephone: 815 395 5678;  Email: [email protected]

Reviewer 2 Report

• There are over four thousand review articles about melanoma targeted treatment, according to https://pubmed.ncbi.nlm.nih.gov/?term=Melanoma%20targeted%20treatment&filter=pubt.review&sort=date I don't see the novelty of this one. To give a couple of examples, Adjuvant and Neoadjuvant Therapies in Cutaneous Melanoma, Oral and Maxillofacial Surgery Clinics of North America, May 2022 provides an overview of immunotherapy treatments for melanoma. Similarly, Potential Biomarkers of Skin Melanoma Resistance to Targeted Therapy-Present State and Perspectives, Cancers, May 2022 provides a review of targeted therapies.

• The main sections headings are BRAF, NRAS, c-KIT, VEGFR. All of these important genes have been known about in the context of melanoma for many years. A review article should aim to summarise very recent findings in the research field with a focus on current developments. The content reads more like an Introduction section of a PhD thesis rather than how a review article should be composed. Are there any contradictory results or controversies in the field? What are the current gaps in knowledge which the recent journal articles were not able to answer?

• Page 1, line 21: "In 2022, it is estimated that 7650 deaths will be attributed to melanoma." Please don't make future predictions in a scientific article. What if a breakthrough new treatment gets FDA approval next month? Extrapolating based on previous data is risky. Instead replace this statement by statistics for calendar year 2021 or earlier.

Author Response

June 23rd, 2022             

To the Editor of Cancers,                                                                                                              

Subject: Submission of revised manuscript No. cancers-1748189: “Treatment of metastatic melanoma with a combination of immunotherapies and molecularly targeted therapies.”

Dear Editor,

We would like to thank the editors and reviewers for their valuable comments and suggestions that have helped in improving the quality of this manuscript. Responses to each comment are provided below. We are confident that this revised manuscript will now be acceptable for publication in Cancers.

Answers to reviewer comments:

Comment;There are over four thousand review articles about melanoma targeted treatment, according to https://pubmed.ncbi.nlm.nih.gov/?term=Melanoma%20targeted%20treatment&filter=pubt.review&sort=date I don't see the novelty of this one. To give a couple of examples, Adjuvant and Neoadjuvant Therapies in Cutaneous Melanoma, Oral and Maxillofacial Surgery Clinics of North America, May 2022 provides an overview of immunotherapy treatments for melanoma. Similarly, Potential Biomarkers of Skin Melanoma Resistance to Targeted Therapy-Present State and Perspectives, Cancers, May 2022 provides a review of targeted therapies.

Response; We agree that targeted and immunotherapies have been reviewed quite extensively on their own, as Reviewer 2 pointed out. Our manuscript, however, goes beyond the individual components and describes drug combinations between targeted and immunotherapies that are in development. Neither of the referenced papers, nor any recent publication describe these combination therapies across pharmaceutical subclasses, as we have attempted to do in this manuscript. The two most recent articles which cover the same premise are included below, from 2011 and 2017, respectively.

PMID: 21847631 – Blank et. al Combination of targeted therapy and immunotherapy in melanoma in Cancer Immunology, Immunotherapy (2011)

PMID: 28114256 – Christiansen et. al. Targeted Therapies in Combination with Immune Therapies for the Treatment of Metastatic Melanoma in The Cancer Journal (2017)

Comment ; The main sections headings are BRAF, NRAS, c-KIT, VEGFR. All of these important genes have been known about in the context of melanoma for many years. A review article should aim to summarise very recent findings in the research field with a focus on current developments. The content reads more like an Introduction section of a PhD thesis rather than how a review article should be composed. Are there any contradictory results or controversies in the field? What are the current gaps in knowledge which the recent journal articles were not able to answer?

Reseponse; In this review we are describing several combinations of targeted and immunotherapy for treatment of melanoma which has been not addressed extensively in earlier papers. Hence additional details have been provided, so that the readers can get a brief background about the individual therapies and understand their combinatory effects. Reducing this content would reduce the clarity of the findings of the review.  

We thank Reviewer 2 for bringing to light the absence of discussion on gaps of knowledge and controversies in the field. The following content has been added in the manuscript to address the reviewer’s concerns.

In BRAF section; Several controversies remain in the field of BRAF-mutated melanoma treatment. Studies have found that 10% of melanoma patients harboring a BRAF mutation have the V600K driver mutation, the second most common type behind V600E. Therapeutic strategies in the context of V600K BRAF-mutated melanoma are not as well elucidated as those with V600E and prognosis is poor. Melanoma driven by V600K mutations is thought to have unique features compared to V600E, including decreased reliance on MAPK/ERK pathway over activation, increased expression of c-KIT, and upregulation of the PIK3CB-AKT anti-apoptotic pathway. Although treatment with previously discussed BRAF/MEK combinatory regimens dabrafenib+trametinib, vemurafenib+cobimetinib, and encorafenib+binimetinib, as well as with immunomodulators like pembrolizumab and iplimumab, has shown some efficacy in V600K-mutant melanoma, evidence is not strong enough to make any formal first-line treatment recommendation for melanoma patients with V600K mutation. Moving forward, the BRAF V600K driver mutation should be independently investigated in greater detail to formulate specified and evidence-backed recommendations of treatments for patients with this mutation, as their prognosis is currently still poor. (Page 8 line 203-218)

In NRAS section; Among the 41 patients enrolled in the phase II dose expansion trial, the median progression-free survival was 3.7 months, and the overall survival was 11.3 months. 20% of the patients achieved a partial response, 51% had stable disease, and 15% had progressive disease (page 11 line 281-287)

To address drug resistance to MEK inhibitor, a preclinical study showed that co-inhibition of MEK and ERK5, a compensatory pathway activated by MEKi possibly via receptor tyro-sine kinase PDGFRβ, suppresses growth and progression of NRAS mutant melanoma cells in vivo and in xenografts [23]. Additionally, inhibition of PDPK1 synergistically en-hances the efficacy of MEKi by stimulating CD8+ T cells [80]. These strategies, though in their early phase of investigation, may help bypass resistance and enhance the efficacy of inhibitors targeting the MAPK pathway. (page 11 line 298-305)

In c-KIT section; Another study assessed therapy with imatinib and NN2101-DM1, a conjugated IgG1 and microtubule inhibitor, both together and alone in c-KIT mutant cancers in mouse models. GIST treated with this combination therapy achieved remission compared to ei-ther therapy alone which highlights a potential avenue of research for combination ther-apy in melanoma management (page 12 line 415-420)

Further, a phase II clinical trial of nilotinib in 25 patients with unresectable melanoma with KIT mutations demonstrated a response rate of 20% and a disease control rate of 56% in patients with exon 11 or 13 mutations after receiving 400 mg oral nilotinib twice daily for 6 months. Of note, the data from this study also showed that STAT3 expression in good responders was significantly decreased, while levels in poor responders did not show significant change. While this clinical trial did not assess outcomes of nilotinib in combination with other therapies, the study showed a significant association between re-duced STAT3 signaling and a clinical response [87]. This study suggests that the JAK/STAT pathway may mediate response to nilotinib and indicates a need to further evaluate the potential for use of JAK/STAT inhibitors in combination with KIT inhibitors in the treatment of KIT-mutated melanoma. Evaluation of combination therapies using KIT inhibitors and STAT inhibitors for treatment of melanoma is thus far minimally stud-ied and therefore demands further consideration. (page 13 line 441-453).

In VEGFR section; However, antiangiogenic administration should be used with caution. Small studies on intravitreal injections of bevacizumab in patients with uveal melanoma and patients with choroidal melanoma resulted in increased tumor growth [121,122]. This phenomenon was also observed after cessation of bevacizumab in colorectal cancer patients, resulting in increased proangiogenic factors and cancer stem cells [123]. Studies also suggests that there is a correlation between ICI therapy, particularly anti-PD-1/PD-L1 antibodies, and hyper progressive disease in a subset of patients with solid tumors [124]. However, another study on melanoma patients treated with ICIs considered hyper progression as a relatively rare phenomenon [125]. These conflicting findings should be considered as potential consequences when combining these therapies (page 14 line 523-532).

Comment; Page 1, line 21: "In 2022, it is estimated that 7650 deaths will be attributed to melanoma." Please don't make future predictions in a scientific article. What if a breakthrough new treatment gets FDA approval next month? Extrapolating based on previous data is risky. Instead replace this statement by statistics for calendar year 2021 or earlier.

Response – After extensive review of the literature, we found that 2021 melanoma case and death statistics are mostly estimates as well. We have updated the statistics for the year 2021.  These estimates of deaths and melanoma cases have also been provided by Cancer statistics 2021 by Jemal et. al (cited 363 times, IF 508) and by Kalyan et. al (PMID – 34698235). These references by Jemal et. al are extensively used in literature and by cancer foundations, including the American Cancer Society, to verify or give approximate statistics of occurrence and death of various types of cancer. Kalyan et. al have published the manuscript in the later part of 2021 and have based their number on SEERs data by NIH.

We hope that these answers suffice the reviewer’s concerns and that the revised manuscript may now be accepted for publication in Cancers.

This manuscript is original and is not under consideration elsewhere. None of the manuscript contents have been previously published. All authors have read and approved all versions of the manuscript and its content for the submission to Cancers.

Sincerely,

Neelu Puri Ph.D. (Corresponding author)Associate ProfessorUniversity of Illinois College of MedicineDept of Biomedical Sciences1601 Parkview AveRockford, IL 61107Telephone: 815 395 5678;  Email: [email protected]

Reviewer 3 Report

This review is focusing on combinational immuno- and targeted molecular therapies of metastatic melanoma. It provides relevant and valuable information on melanoma treatment options not only for the clinic but also for researchers to understand advantages and disadvantages of particular combinations that may advance development of new therapeutic strategies. The review is very well written and except of some tipos to be corrected it is ready for publication. Therefore, I recommend the present paper for acceptance and its consequent publication in Cancers.

Author Response

February 23rd, 2021    

To the Editor of Cancers,           

Subject: Submission of revised manuscript No. cancers-1748189: “Treatment of metastatic melanoma with a combination of immunotherapies and molecularly targeted therapies.”

Dear Editor,

We would like to thank the editors and reviewers for their valuable comments and suggestions that have helped in improving the quality of this manuscript. Responses to each comment are provided below. We are confident that this revised manuscript will now be acceptable for publication in Cancers.

Answers to reviewer comments:

Comment; This review is focusing on combinational immuno- and targeted molecular therapies of metastatic melanoma. It provides relevant and valuable information on melanoma treatment options not only for the clinic but also for researchers to understand advantages and disadvantages of particular combinations that may advance development of new therapeutic strategies. The review is very well written and except of some tipos to be corrected it is ready for publication. Therefore, I recommend the present paper for acceptance and its consequent publication in Cancers.

Response; We are glad to know that the reviewer is happy with the contents of the manuscript, particularly advantages and disadvantages of combinatory treatments that may advance development of new therapeutic strategies.  As suggested by the reviewer, we have corrected any typos we could find in the manuscript.

We hope that these answers suffice the reviewer’s concerns and that the revised manuscript may now be accepted for publication in Cancers.

This manuscript is original and is not under consideration elsewhere. None of the manuscript contents have been previously published. All authors have read and approved all versions of the manuscript and its content for the submission to Cancers.

Sincerely,

Neelu Puri Ph.D. (Corresponding author)Associate ProfessorUniversity of Illinois College of MedicineDept of Biomedical Sciences1601 Parkview AveRockford, IL 61107 Telephone: 815 395 5678;  Email: [email protected]

Round 2

Reviewer 2 Report

The researchers have addressed each of my concerns in turn. Nice to see discussion of the caveats, such as V600K and hyperprogression.

Author Response

July 2nd, 2022

To the Editor of Cancers,

Subject: Submission of revised manuscript No. cancers-1748189: “Treatment of metastatic melanoma with a combination of immunotherapies and molecularly targeted therapies.”

Dear Editor,

We would like to thank the editors and reviewers for their valuable comments and suggestions that have helped in improving the quality of this manuscript. Responses to each comment are provided below. We are confident that this revised manuscript will now be acceptable for publication in Cancers.

Answers to reviewer 2 comments:

Comment; The researchers have addressed each of my concerns in turn. Nice to see discussion of the caveats, such as V600K and hyperprogression.

Response; We are glad to know that the reviewer is happy with the addressed comments of the manuscript, particularly the discussion of the caveats, such as V600K and hyperprogression. As suggested by the reviewer, we have corrected any typos and grammar we could find in the manuscript.

We hope that these answers suffice the reviewer’s concerns and that the revised manuscript may now be accepted for publication in Cancers.

This manuscript is original and is not under consideration elsewhere. None of the manuscript contents have been previously published. All authors have read and approved all versions of the manuscript and its content for the submission to Cancers.

Sincerely,

Neelu Puri Ph.D. (Corresponding author)

Associate Professor

University of Illinois College of Medicine

Dept of Biomedical Sciences

1601 Parkview AveRockford, IL 61107

Telephone: 815 395 5678; 
